# The Claiming Costs Scale: A new instrument for measuring the costs potential beneficiaries face when claiming social benefits

Julie Janssens[1]*, Tim Goedemé[1,2], Koen Ponnet[3]

**1** Department of Sociology, Herman Deleeck Centre for Social Policy, University of Antwerp, Antwerp, Belgium, **2** Department of Social Policy and Intervention, Institute for New Economic Thinking at the Oxford Martin School, University of Oxford, Associate Member of Nuffield College, Oxford, United Kingdom, **3** Faculty of Social Sciences, imec-mict-Ghent University, Ghent, Belgium

\* julie.janssens@uantwerpen.be

**Data Availability Statement:** All data files are available from the openICPSR database (accession: https://doi.org/10.3886/E131962V3). The full questionnaire and datasets are part of a larger database including variables provided by the

## Abstract

It is a well-known feature of social protection systems that not all persons who are entitled to social benefits also claim these benefits. The costs people face when claiming benefits is considered an important cause of this phenomenon of non-take-up. In this paper, we developed and examined the psychometric properties of a new scale, the Claiming Cost Scale (CCS), which measures three dimensions of costs associated with claiming benefits. A multi-phase instrument development method was performed to develop the instrument. The item pool was generated based on a literature review, and presented to academic experts (*n* = 9) and experts by experience (*n* = 5) to assess content and face validity. In a second stage, centrality and dispersion, construct validity, convergent and divergent validity, and internal reliability of the instrument were tested. These analyses were based on two samples (*n* = 141 and *n* = 1265) of individuals living in low-income households in Belgium. Nine items were retained, which represent three factors (Information costs, Process costs and Stigma). The confirmatory factor analysis proved adequate model fitness. Both convergent and divergent validity were good, and internal consistency was adequate, with Cronbach's alpha ranging between .73 and .87. The findings showed that the CCS is a valid and reliable instrument for assessing the costs potential beneficiaries face when claiming benefits. Consisting of only nine items, the scale can be easily implemented in large-scale survey research or used in day-to-day work of service providers who are interested in understanding non-take-up of their service.

## Introduction

Social benefits and services aimed at supporting people in vulnerable situations, including minimum income support, social housing and housing benefits, subsidized health insurance, family benefits, old-age support and unemployment benefits, are an important part of modern

Belgian Crossroads Bank for Social Security (CBSS) and Belgian tax authorities, which requires a specific authorization of the Belgian Data Protection Authority. Researchers who are interested in the full dataset and questionnaires can contact Julie Janssens (Julie.Janssens@uatwerpen.be) or Tim Goedemé (tim.goedeme@spi.ox.ac.uk).

**Funding:** This work was funded by the Belgian Federal Science Policy Office (belspo) under the BRAIN-be program [number BR/154/A4/TAKE]. The funders had no role in study design, data collection and analysis, decision to publish, or preparation of the manuscript.

**Competing interests:** The authors have declared that no competing interests exist.

welfare states. However, empirical evidence shows that in many European countries only a fraction of those that are supposed to benefit, receive the benefits to which they are entitled [1,2]. From a policy perspective, this phenomenon of non-take-up has been considered both a strength and a flaw of social policies: whereas some claim it helps to make targeting of benefits more efficient if non-take-up results in benefit receipt only by those most in need [3]; others have pointed to losses in efficiency and effectiveness [2,4,5], reduced equity in access to benefits [4,6,7]; and potential longer-term negative budgetary consequences [1,8]. From a scientific perspective, non-take-up is an intriguing phenomenon as at first sight, one would expect that households would take up their social rights, given that it would benefit them financially. To explain non-take-up, researchers and policymakers have long believed that the expected, perceived and experienced costs (potential) beneficiaries face when claiming benefits play an important role in explaining the non-use of benefits [7,9]. Still, at the moment, a comprehensive and validated instrument to measure the costs associated with claiming benefits in a more direct way is lacking. Therefore, the first aim of this study is to develop an easy to administer instrument, called the Claiming Costs Scale (CCS), that measures in a direct way various dimensions of the costs that potential and actual beneficiaries face when applying for benefits. The second aim of this study is to test the psychometric properties of the CCS in terms of validity and reliability. In case of sound psychometric properties, the CCS can be used by researchers, service providers, and policy makers to (1) gain a better understanding of the reasons why people forgo the benefits they are entitled to and (2) to identify interventions that reduce claiming costs in an efficient way.

## Claiming costs

The costs associated with claiming benefits are commonly grouped into three main categories: i) information costs, ii) process costs and iii) social and psychological costs (including stigma) [7,10–12]. This taxonomy is however not exhaustive and other classifications have been proposed [13]. In what follows we define each of these dimensions of claiming costs and summarize how they have been measured in the past.

**Information costs.** Basic information on the existence of social benefits, their eligibility conditions and the application procedure are necessary for potential claimants to become aware of their eligibility and eventually file an application. Information costs can be defined as the expected, perceived and experienced time and effort that people have to invest into gathering and understanding information on the existence of public provisions, the eligibility criteria, the claiming process and its consequences (e.g. the implications of a successful claim for entitlement to other benefits) [14]. In this sense information costs refer to the role of imperfect information, misinformation or a lack of information as this can increase information costs for potential claimants, i.e. by making it harder to be correctly informed about the benefit and its application procedure [14]. There are three ways in which the role of information (costs) is measured in quantitative and qualitative research: (1) indirectly, by using socio-demographic proxies for the level of information costs, attributing a higher level of non-take-up among particular socio-demographic groups (for example lower-educated households or immigrants) as these groups are assumed to face higher costs [15,16]; (2) more directly, by asking respondents about the main reasons for not applying for benefits in surveys [17,18] or by asking them specifically about their knowledge of certain benefits [19–21]; and (3) directly, by evaluating the effect of interventions that reduce information costs, such as sending letters or flyers to a group of potential beneficiaries and compare the effect on benefit take-up with a control group (with similar characteristics) that did not receive the intervention [22–25].

**Process costs.** Process costs, also known as administrative costs, application costs or transaction costs, refer to the expected, perceived and experienced time, money and energy spent in the claiming process itself [14]. One can think of difficulties with and the time and energy spent on completing forms, providing documentation, travel costs or time lost with queuing [12]. Also process costs are commonly identified in three different ways: (1) indirectly, most often by using the same proxies as the ones used for identifying information costs (i.e. educational level, immigrant status, area of residence or household composition) [e.g. 26–28] as the distinction between information and process costs is not always recognised or well-defined in the literature [e.g. 29–31]; (2) more directly, by asking in a survey to indicate the main reason for not applying or prematurely stopping the claiming process [e.g. 20,32]; and (3) directly, by making use of a field experiment [e.g. 33].

**Social and psychological costs.** The last category of costs refers to social and psychological costs. Social and psychological costs may include social exclusion (including weakening of friendships and family ties), experience of humiliation or negative treatments, loss of self-esteem, and loss of social status. When discussing the psychological and social costs related to the take-up of public programmes, academics mainly focus on stigma resulting from being "on welfare" [34]. A widespread definition for stigma is that "stigmatized individuals possess (or are believed to possess) some attribute, or characteristic, that conveys a social identity that is devalued in a particular social context" [35]. In this case, we study stigma in relation to the attribute of claiming or receiving benefits [cf. 36]. Although stigma in relation to non-take-up has also been studied on the basis of proxies [e.g. 26,27,37], there are efforts to measure stigma associated with claiming benefits more directly [19,36,38,39]. We have built on this research to select the survey questions to develop the claiming costs scale.

In sum, to the best of our knowledge there is a lack of a comprehensive and standardized instrument which measures the different categories of claiming costs in a more direct way. The commonly used proxy-based approach can only provide some indirect evidence about the relative importance of these costs and does not allow for making a distinction between the various dimensions of claiming costs. More direct ways of measuring, such as conducting field experiments or randomised control trials are time- and cost-intensive and often not feasible in practice. In addition, we do not know of a measure that assesses the different types of claiming costs with one single instrument. Furthermore, the relevance of using brief and reliable self-report measures to operationalize social, psychological and behavioural constructs has been emphasized [40]. Therefore, the current study aims to develop a short and reliable self-report instrument that captures the aforementioned types of costs associated with claiming benefits with a single scale.

As claiming costs can be expected to be most relevant for non-take-up in the case of means-tested benefits, we developed the Claiming Costs Scale in the context of a study of social assistance benefits, focusing on benefits for able-bodied persons at active age living in Belgium. In European welfare states, social assistance benefits serve as a final safety net for those who live in precarious conditions but fall through the cracks of social security [e.g. 41,42]. Social assistance benefits are fully financed with government resources and their goal is to provide a minimum level of income support to individuals and households living in poverty, which should enable their integration in society. Likewise, in Belgium, individuals who are eligible for social assistance receive a monetary benefit which tops their incomes up to a pre-defined threshold. Currently, 1.4 percentage (approximately 146,800 persons) of the total population in Belgium receives a social assistance benefit for individuals at active age. In order to apply for this benefit, a claim has to be filed with the Public Centre for Social Welfare (PCSW) of the municipality in which one lives. During the application process, a social worker carries out a social investigation, which includes a means-test, to determine the applicant's eligibility. Non-take-up of

means-tested social assistance benefits are found to be particularly high. Although precise estimates are lacking, non-take-up for the social assistance benefit for able-bodied has been estimated to be almost 60% in Belgium [43].

## Materials and methods

The study was conducted in three phases, following the recommended steps for scale development [44], and consistent with recent articles in which the psychometric properties of instruments were tested [45–47]. In the first phase, we started with item generation and scale development by stating the concepts we wanted to measure (i.e., information costs, process costs and stigma), generating candidate items for the different concepts, determining the most appropriate phrasing for each item, and assessing the face and content validity of the items in a qualitative way. The second and third phase were the testing and evaluation phase respectively, involving two cross-sectional samples ($n_{sample1}$ = 141 and $n_{sample2}$ = 1265) of people living in low-income households. We carried out exploratory and confirmatory factor analysis in order to examine the construct validity. Furthermore, our instrument was evaluated in terms of convergent and divergent validity, and reliability. Below, we elaborate in detail on the different stages by discussing the statistical analyses performed and data used at each stage of the scale development process. The study protocol was approved by the Ethics Committee for Social Sciences and Humanities of the University of Antwerp (final positive clearance for SHW_18_69 (sample 1) and SHW_18_32 (sample 2)) and informed consent was obtained from all participants by means of a registered question at the beginning of the questionnaire. All data were analysed anonymously, and cannot be linked back to identifying information.

### Phase 1: Item development

**Item generation.**   To generate candidate items for our Claiming Costs Scale a deductive approach was used. First, we examined the literature on non-take-up and searched for previous studies that measured the costs associated with claiming benefits. In particular, items from two Dutch questionnaires on the non-take-up of benefits [19,39] were taken as starting point. For items related to stigma we drew also inspiration from the Stigma Scale developed by King, Dinos [48], a standardized scale to measure stigma associated with having a mental illness. When necessary, items were reformulated and adapted to the context of claiming social assistance benefits. Lastly, to capture aspects that were not (adequately) covered within existing questionnaires, new items were generated based on academic and policy-related literature and researcher intuition. Subsequently, a team consisting of five researchers screened the items for representativeness and redundancy. Items deemed redundant or non-representative were omitted after discussion.

**Content validity and face validity.**   Next, the remaining items were evaluated on content validity and face validity in order to assess whether the set of items comprehensively covered the different concepts we aimed to measure; and whether the items appeared appropriate, relevant and clear to our target population [44].

*Content validity*. Content validity was assessed in a qualitative way, by consulting a panel of academic and non-academic researchers specialized in surveying low-income groups and measuring poverty (*n* = 9). The expert panel was asked to assess the relevance, the wording, the grammar, item allocation, and scaling of the items as part of a more extensive questionnaire on non-take-up of social benefits in Belgium.

*Face validity*. Face validity was examined by asking a group of people who formerly have lived in severe poverty and deprivation ("experts by experience") (*n* = 5) to evaluate the same questionnaire as presented to the experts. The experts by experience were recruited with the

help of the Belgian Federal Public Planning Service Social Integration, which is a government organisation that handles the implementation, assessment and monitoring of social integration policy in Belgium. Participants were sent the questionnaire beforehand and the discussion of the items took the format of a focus group in presence of the first author. The participants were asked whether the items related to the costs associated with claiming benefits, whether items were missing or redundant and whether the items were clear and easy to understand.

## Phase 2: Scale development

**Survey administration (Sample 1).**   In order to assess construct validity using exploratory factor analysis (EFA), the potential scale items were administered in an online survey among a sample of individuals living on a relatively low income in Belgium (Sample 1) in September 2017. We defined low-income households as having a total disposable household income of less than 20,000 EUR a year for households with one adult, and less than 35,000 EUR a year for households including more than one adult. Respondents were recruited from both Dutch and French speaking parts of Belgium. We used a quota sampling design to ensure an equal representation of Dutch and French speaking respondents. Participants were recruited in collaboration with the digital data collection provider "Research Now", from their market research panels. All Research Now research activities are based on informed consent ("opt-in"). Once new panel members register, the information provided at that point is very clear on the purpose of their membership and they are required to complete the panel registration and agree to all terms and conditions upon registration. In addition, each invited panel member was free to participate in our study and could simply refuse not to participate to the online survey that was sent to them by e-mail. In addition to the claiming costs items, the survey also collected information on participants' experience with applying for benefits, their income situation and socio-demographic characteristics. The survey questionnaire translated into English is provided in S1 Appendix. The completion of the questionnaire lasted on average 27 minutes. Respondents who took part in the survey were rewarded after completing the survey according to a structured incentive scheme. The items of the preliminary CCS version were asked to 141 respondents between 18 and 65 years old who indicated to know the social assistance benefit. Descriptive statistics of this sample can be found in Table 1 (Sample 1).

**Item reduction.**   Using Sample 1, we evaluated the performance of the individual items in order to identify the appropriate ones to constitute the scale and to drop the items that are not or are the least-related to the concept(s) we try to measure [40]. First, we conducted a basic univariate analysis to evaluate the normality, means and standard deviations of the items. As recommended by DeVellis [40] items with a mean close to the centre of the range of possible scores and items with a relatively high variance are usually preferred. In addition, we calculated the corrected item-total correlations to ensure a set of items that are highly intercorrelated [40]. Items without adequate variance and low item-total correlations ($< 0.3$) were identified for deletion [49].

**Exploratory factor analysis (EFA).**   EFA was applied to identify the main factors of our instrument and to retain a parsimonious set of items. As recommended by Gable and Wolf [50] a sample of at least 5 to 10 participants per item is required to ensure a theoretically clear factor structure for EFA. With 141 participants, we fall within this minimum range. To evaluate the adequacy of our sample for factor analysis the Kaiser-Meyer-Olkin (KMO) test and Bartlett's test of sphericity were applied [51]. KMO values of .60 or higher indicate an acceptable sample and P-values of less than .05 indicate that factor analysis is useful to apply on the data [45]. We performed an EFA using principal axis factoring and varimax rotation to extract the main factors. Any factor with an eigenvalue above 1 was considered significant for factor

**Table 1. Characteristics of the study samples.**

| | Sample 1 (n = 141) Number (%) (n = 141) | Sample 2 (n = 1265) Number (%) |
|---|---|---|
| **Age** | | |
| 18–34 years | 42 (30%) | 317 (25%) |
| 35–49 years | 43 (30%) | 458 (36%) |
| 50–64 years | 56 (40%) | 416 (33%) |
| +65 | 0 (0%) | 74 (6%) |
| **Gender** | | |
| Male | 55 (39%) | 517 (41%) |
| Female | 86 (61%) | 732 (59%) |
| **Household type** | | |
| Single person household | 6 (4%) | 356 (28%) |
| Two-person household | 65 (46%) | 277 (22%) |
| Three-person household | 34 (24%) | 223 (18%) |
| Four-person household | 17 (12%) | 181 (14%) |
| +4-person household | 19 (14%) | 228 (18%) |
| **Education** | | |
| No degree | 5 (4%) | 66 (5%) |
| Primary education | 13 (9%) | 152 (12%) |
| Secondary education | 87 (62%) | 758 (60%) |
| Higher education | 36 (25%) | 250 (20%) |
| **Activity status** | | |
| Employed | 56 (40%) | 416 (33%) |
| Unemployed | 16 (16%) | 272 (22%) |
| Disabled | 25 (18%) | 212 (17%) |
| Retired | 20 (14%) | 92 (7%) |
| Other (student, homemaker) | 17 (12%) | 267 (21%) |
| **Region** | | |
| Flemish Region | 72 (51%) | 620 (49%) |
| Walloon Region | 57 (40%) | 314 (25%) |
| Brussels-Capital Region | 12 (9%) | 329 (26%) |
| **Monthly disp. income** | | |
| Less than 499,99 EUR | 14 (10%) | 28 (2%) |
| 500–1499,99 EUR | 43 (30%) | 525 (41%) |
| 1500–1999,99 EUR | 25 (18%) | 240 (19%) |
| 2000–2499,99 EUR | 35 (25%) | 161 (13%) |
| More than 2500 EUR mmomonth | 24 (17%) | 311 (25%) |
| **Able to ends meet** | | |
| (very) difficult | 42 (30%) | 726 (57%) |
| rather difficult | 58 (41%) | 236 (19%) |
| rather easy | 32 (23%) | 199 (16%) |
| (very) easy | 9 (6%) | 101 (8%) |

extraction, and a scree plot was used to specify the number of factors. Factor loadings equal to or greater than .40 were considered acceptable [52], however factor loadings greater than .50 were preferred [53,54]. To ensure that only parsimonious, functional and internally consistent items are included in the final instrument [44], we used the following procedure to identify further items for deletion. First, items which loaded on two distinct factors (cross loadings of

above .32) were dropped [55]. Second, items with factor loadings below .50 were deleted as well [54].

### Phase 3: Scale evaluation

**Survey administration (Sample 2).** As recommended by DeVellis [40] and Dimitrov [56] we re-administered the scale items in a different, representative sample of the target population in order to validate the results obtained in the first sample and to (further) evaluate the psychometric properties of our new scale by means of confirmatory factor analysis. The data we used were collected in the context of the TAKE Project (https://takeproject.wordpress.com), a network research project between two universities and two public institutions, in which one of the main activities is the organization of a representative survey among the Belgian population with a low income to study the non-take-up of a wide range of Belgian public policy provisions. To select the sample for this survey, we used a two-stage stratified sample design with random sampling at both stages. At the first stage a sample of municipalities was selected, with probabilities of selection proportional to estimated size, stratified according to geographical characteristics and the degree to which persons entitled to social assistance are overrepresented in the population. At the second stage, a sample of households was selected without replacement for each selected municipality, stratified by age and social assistance status. The sampling frame consisted of national register data, enriched with data on income and benefit receipt from the tax and social security registers. Next, sample members were sent a letter with an explanation about the study and an invitation to participate. Sample members were assured they could refuse at any time to participate in the study. Respondents that agreed to participate, were interviewed by trained interviewers by means of computer-assisted face-to-face interviews. At the start of the interview, respondents were asked to give their informed consent to participate in the survey by means of a registered question at the beginning of the questionnaire. If the respondent refused to agree with the conditions of the study at this specific moment, the interview was immediately interrupted. Interviews were conducted between September 2019 and September 2020, with a break between March and August 2020 due to Covid-19. In total, almost 2000 households living in the three regions of Belgium were interviewed. For this study, we made use of the information collected from the household head, who provided detailed information on their knowledge, experience and receipt of benefits, their financial situation, sociodemographic characteristics, household composition, housing situation and competences. Respondents who indicated to know the social assistance benefit for persons at active age and were between 18 and 64 years old, were asked to respond to the items of the Claiming Costs Scale. In total, 1,265 household heads responded to the items of the Claiming Costs Scale. The descriptive characteristics of this group (Sample 2) can be found in Table 1. The questions used for this study are translated into English and are provided in S2 Appendix.

**Confirmatory factor analysis.** In order to validate the theoretical (factor) structure derived from the EFA in Sample 1, a test of scale dimensionality was conducted on Sample 2 using confirmatory factor analysis (CFA) with Maximum likelihood estimation. To evaluate the fit of the model, we used several fit criteria, including relative Chi-square, Comparative Fit Index (CFI), Tucker Lewis Index (TLI), Root Mean Square Error of Approximation (RMSEA) and Standardized Root Mean Square Residual (SRMR) [57,58]. We use the relative Chi-square, which equals the Chi-square value divided by the degrees of freedom, as this value might be less sensitive to sample size then the classic Chi-square index. A reasonably well-fitting model should have a statistically insignificant Chi-square, however in larger samples (n>200) the model will be routinely rejected based on the Chi-square value. Despite the fact there is still

some discussion about the value the relative Chi-square should have (for an overview see [59]), most researchers recommend to have a value of 5 or less. For both TLI and CFI, values close to .95 or greater indicate a well-fitting model, whereas values in the range of .90 and .95 may be indicative of acceptable fit [58]. For RMSEA, values between .08 and .10 suggest an average model fit, between .06 and .08 an adequate fit and values of less than .05 suggest good model fit [60]. For SRMR values close to .08 or below and ideally less than .05 are usually seen as indicating a reasonably good and good model-data fit, respectively [58,61].

**Reliability.**   Cronbach's alpha coefficient was used to measure the reliability of the whole scale, as well as for each subscale separately. Alpha values of at least .70 are often considered to be satisfactory, but values above .80 are preferred as they indicate a very good internal consistency [62].

**Validity.**   *Item-convergent validity*. Item-convergent validity was assessed by examining the correlations between the item scores and subscale scores of the CCS by use of Spearman's correlation coefficient. Item-convergent validity is met when for each subscale of the CCS, each item of the subscale significantly correlates more with the total score of its respective subscale, rather than with the total score of other subscales [45]. Correlation coefficient values between 0 and .20 are considered poor; between .21 and .40, fair; between .41 and .60, good; between .61 and .80, very good; and above .81, excellent [63].

*Convergent and discriminant validity*. To establish convergent and discriminant validity of each subscale, we examined the factor loadings of the items and the average variance extracted (AVE) of the different dimensions (factors), which is the average amount of variance in the items that a latent factor can explain [64]. Following the criteria of Fornell and Larcker [64] and Hair Jr, Black [54], convergent validity is demonstrated when all items have significant factor loadings on the respective latent factor ($p < .001$) and the AVE for each factor is greater than .50. Next, discriminant validity is evaluated by demonstrating that the AVE for each dimension is greater than the squared correlation between it and any other dimension of the construct [64]. If this is the case, it shows that each dimension explains better the variance of its own indicators than the variance of the other dimensions [65].

## Results

### Phase 1: Item development

**Item generation.**   Based on a literature review, the item generation process resulted in 40 items associated with three key dimensions of claiming costs: information costs, process costs and stigma. The 40 items and item sources can be provided upon request. After consulting an expert panel (five scientific researchers acquainted with the topic) about the relevance of the items, 15 items were retained. Subsequently, the 15 items were translated from English to Dutch and French (i.e. the two most spoken languages in Belgium). Translation to both languages was performed by the research team and a team of certified translators checked for differences between the Dutch, French and English versions and ensured equivalence of item formulation across languages.

**Content and face validity.**   The 15-item scale was reviewed by a panel of experts and a group of persons that have experienced severe poverty and deprivation ("experts by experience") in order to examine the content and face validity of the scale, respectively. Based on the feedback of the experts, we adapted the wording of the stigma items in a way that the questions sounded logical for both respondents that received the particular benefits and respondents that did not (already) receive the benefits but who might be entitled to them. This is important, as it allows for comparing claiming costs among both those who take up the benefit concerned and those who are eligible but do not take up the benefit. For instance, "people I see regularly,

**Table 2. Items, mean, standard deviation, item-to-total correlation and item source.**

| Construct | Item wording | Mean | SD | Item Total Correlation | Source |
|---|---|---|---|---|---|
| **Information Costs** | | | | | |
| IC1 | I know the benefits of the social assistance benefit | 2.66 | 1.16 | 0.30 | Research team |
| IC2 | I have a fairly good idea whether I am entitled to a social assistance benefit | 2.43 | 1.13 | 0.36 | Research team |
| IC3 | I find it easy to find all the necessary information on the social assistance benefit | 2.55 | 0.99 | 0.23 | Research team |
| IC4 | I know the procedure for applying for a social assistance benefit | 2.81 | 1.23 | 0.33 | Research team |
| **Process Costs** | | | | | |
| PC1 | It is a lot of work to apply for a social assistance benefit | 2.72 | 0.94 | 0.59 | Research team |
| PC2 | The procedure for applying for a social assistance benefit is difficult | 2.73 | 0.95 | 0.57 | Research team |
| PC3 | People have to answer a lot of intrusive and personal questions while applying for a social assistance benefit | 2.38 | 1.00 | 0.51 | Research team |
| PC4 | It is difficult for me to go to the Public Centre for Social Welfare* during opening hours to apply for a social assistance benefit | 2.98 | 1.10 | 0.33 | Stuber et al., 2004** |
| PC5 | All things considered, it takes a lot of time to claim a social assistance benefit | 2.62 | 0.93 | 0.57 | Research team |
| **Stigma** | | | | | |
| S1 | It is better that other people do not know if you receive a social assistance benefit | 2.54 | 1.09 | 0.59 | Van Oorschot, 1994 |
| S2 | If someone receives a social assistance benefit he or she should be ashamed | 3.71 | 1.09 | 0.41 | Research team |
| S3 | People I see regularly, would look down on me if I would receive a social assistance benefit | 3.00 | 1.18 | 0.47 | Wildeboer et al., 2007 |
| S4 | When I would receive a social assistance benefit, this would give me the feeling that I'm begging | 2.79 | 1.17 | 0.45 | Wildeboer et al., 2007 |
| S5 | The society is not understanding towards people who are receiving a social assistance benefit | 2.29 | 0.96 | 0.60 | King et al., 2007** |
| S6 | If I would receive a social assistance benefit, I would be ashamed | 2.93 | 1.24 | 0.41 | King et al., 2007** |

*The public administration where a claim for social assistance should be filed.

**Items are modified to the specific context of claiming a social assistance benefit.

IC: Information costs, PC: Process costs, S: Stigma.

look down on me when I receive a social assistance benefit" was changed to "People I see regularly, *would* look down on me if I *would* receive a social assistance benefit". In addition, we changed the wording of some other items that the experts found confusing, unclear or likely to yield ambiguous responses. At this stage, no additional items were added or deleted. We further finetuned some of the items, after having received the (minor) comments of the experts by experience, while no more items were eliminated or added. Therefore, the pre-final instrument consisted of 15 items, covering the three types of costs: information costs (items 1–4), process costs (items 5–9) and stigma (items 10–15). Answers ranged from 1 = *strongly agree* to 5 = *strongly disagree*. Table 2 lists the 15 items and their sources.

## Phase 2: Scale development

**Item reduction.** After administering the items in Sample 1, univariate analyses of the 15 items yielded acceptable results: items had means ranging from 2.29 to 3.71 on a 5-point scale with standard deviations ranging from 0.93 to 1.24, and there were no items with responses concentrated in only one or two response categories. The item IC3 had however a weak item-total correlation coefficient (<0.3), and was therefore dropped. Table 2 provides an overview of the descriptives.

**Exploratory factor analysis.** We performed an exploratory factor analysis (EFA) with varimax rotation on Sample 1 for the remaining 14 items. The KMO was .769 and the Barlett's

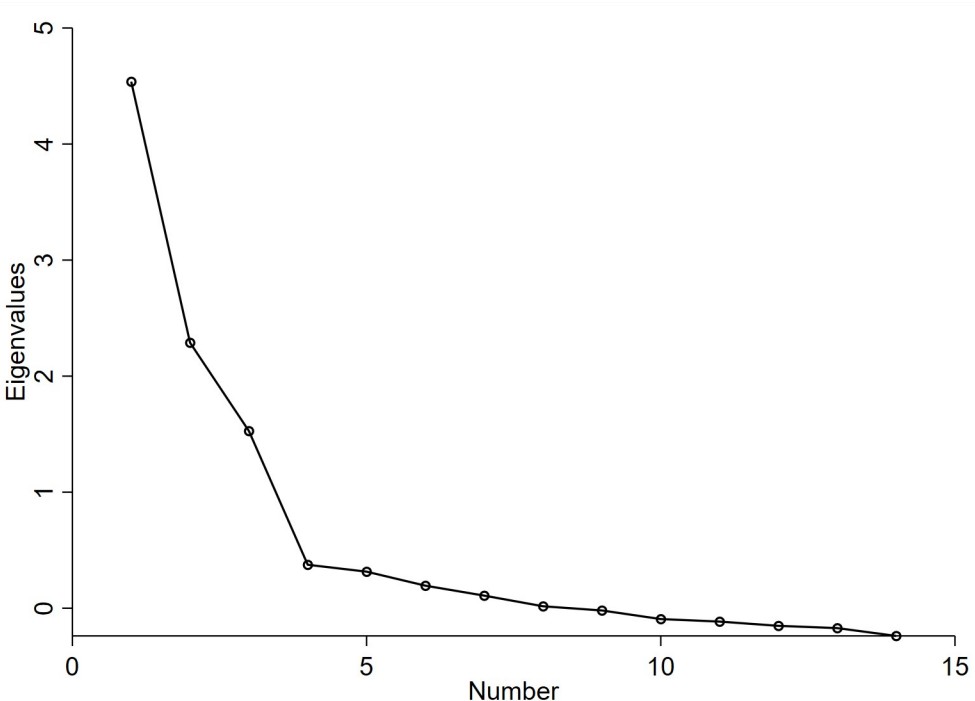

**Fig 1. Scree plot for determining factors of the designed instrument (Sample 1, n = 141).**

test of sphericity was significant ($\chi$2(91) = 1155.903, $p$ < .001) indicating that our sample was adequate for EFA. Initially, for the 14-item instrument, three factors showed eigenvalues greater than 1, explaining 59.78% of observed variance, while the scree plot showed a four-factor solution (Fig 1). This factor solution was further explored by repeatedly assessing the item performance of the items. Eventually, based on the assessment of the item and cross loadings, four items (S1, S5, PC3, PC4) showed factor loadings below .50 and therefore were step-by-step removed using iterative factor analysis. This process resulted in a good three-factor solution for ten items explaining 67.72% of variance. The three factors were: Factor 1 (Stigma) including four items and accounting for 32.05% of the explained variance, Factor 2 (Process costs) including three items explaining 21.35% of the observed variance, and Factor 3 (Information costs) including three items and explaining 14.31% of the observed variance. Table 3 shows the factor loadings of the items.

## Phase 3: Scale evaluation

**Confirmatory factor analysis.** We conducted a confirmatory factor analysis with a different sample (Sample 2) to examine whether the hypothesized three-factor structure obtained from the exploratory factor analysis was a good fit for the data. As our data were non-normally distributed, results were obtained with maximum likelihood mean adjusted. The initial model provided an adequate fit: $\chi^2$(32) = 143.40, p < .001, $\chi^2$/df = 4.48, CFI = .97, TLI = .96, SRMR = .04, RMSEA = .05 (CI:.04 - .06). All standardized factor loadings were statistically significant at the 0.001 level and above .50 except for item 'S3' (item loading of .38). After deleting this item, all fit indices indicated an adequate to good fit: $\chi^2$(24) = 96.71, p < .001, $\chi^2$/df = 4.03, CFI = .98, TLI = .97, SRMR = .03, RMSEA = .05 (CI: .04 - .06). All factor loadings were .49 or above. Only the correlation between the latent factor 'information costs' and 'process costs' was

**Table 3. Exploratory factory analysis of the claiming costs scale– 10 items (Sample 1, n = 141).**

| Item | Factor 1 Stigma | Factor 2 Process costs | Factor 3 Information costs |
|---|---|---|---|
| S6 | **0.852** | 0.112 | -0.121 |
| S4 | **0.822** | 0.150 | -0.108 |
| S3 | **0.750** | 0.140 | -0.053 |
| S2 | **0.646** | 0.097 | 0.116 |
| PC1 | 0.164 | **0.942** | 0.050 |
| PC2 | 0.184 | **0.893** | 0.039 |
| PC5 | 0.116 | **0.781** | 0.068 |
| IC1 | -0.057 | 0.053 | **0.832** |
| IC4 | -0.060 | 0.050 | **0.792** |
| IC2 | 0.006 | 0.036 | **0.757** |

Note: Highest factor loading per item in bold.

IC: Information costs, PC: Process costs, S: Stigma.

positively significant, indicating that a higher degree of information costs is also associated with a higher degree of process costs. In contrast, we did not find a significant relation between the degree of stigma and degree of information or process costs. Fig 2 shows the model with the standardized factor loadings of the items and correlation coefficients between the latent factors.

**Reliability.** In order to evaluate the instrument's reliability, Cronbach's alpha was calculated separately for the whole scale and each subscale separately. Cronbach's alpha coefficient for the total scale was .73, and for the subscales Information costs, Process costs and Stigma respectively .75, .87 and .73 indicating satisfactory to good internal consistency.

**Validity.** *Item-convergent validity*. Item-convergent validity for the Claiming Cost Scale is presented in Table 4. As can be seen, all coefficients are higher than .70 with most of them

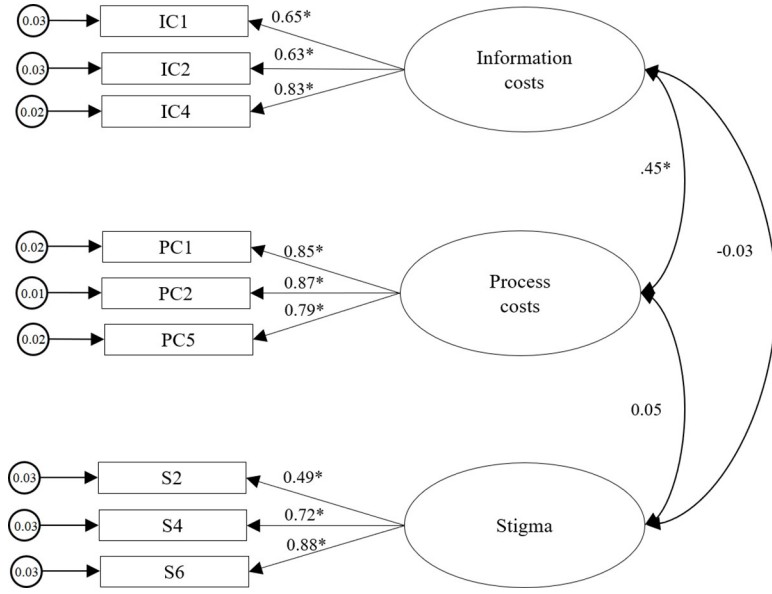

**Fig 2. The 3-factor model for the CCS obtained from confirmatory factor analysis (Sample 2, n = 1265).** Note. *p < .001.

**Table 4. Item-convergent validity: Item-scale correlation matrix for the three subscales of the CCS (Sample 2, n = 1265).**

| Items | Information costs | Process costs | Stigma |
|---|---|---|---|
| IC1 | **0.81** | 0.25 | -0.00 |
| IC2 | **0.81** | 0.18 | 0.00 |
| IC4 | **0.83** | 0.40 | -0.01 |
| PC1 | 0.32 | **0.90** | 0.04 |
| PC2 | 0.27 | **0.91** | 0.07 |
| PC5 | 0.32 | **0.88** | 0.05 |
| S4 | 0.00 | 0.06 | **0.84** |
| S6 | -0.03 | 0.02 | **0.86** |
| S2 | 0.03 | 0.07 | **0.70** |

above .81, indicating very good to excellent correlation coefficients and demonstrating all items to be correlated with their own hypothesized subscales. The subscale Process costs had the highest item-convergent validity.

*Convergent and divergent validity.* For convergent validity, all standardized factor loadings should be above .50 and AVE for each subscale should be greater than .50 [54,64]. Fig 2 shows all items to have positive standardized factor loadings above .50, except for item 'S2' (item loading of .49). In addition, AVE values for the three subscales were above .50, lending further support for the convergent validity of the scale (Table 5). Finally, comparing the variance extracted estimates (i.e. AVE) of the subscales with the shared variance (i.e. the squared correlations) between the subscales, we found that all AVE values were greater than the squared inter-subscale correlations, which supports the discriminant validity between the subscales (Table 5).

## Discussion

A substantial body of literature suggest that the costs (potential) beneficiaries face when claiming social benefits are among the most important reasons why people do not take up the benefits they are entitled to [7,9]. Despite the relevance of these costs to understand the problem of non-take-up, to our knowledge there is no validated measure that assesses both the practical, social and psychological costs (potential) beneficiaries experience or perceive when claiming benefits. Therefore, the current study set out to develop a short and reliable self-report instrument to measure different dimensions of costs associated with claiming benefits: the Claiming Costs Scale (CCS). The CCS items were designed to assess three key dimensions of claiming costs: information costs, process costs and stigma. To develop the scale, the recommended steps for the creation of a new scale were followed, in accordance with prevailing standards [44]. Statistical analyses were conducted on two samples of persons living in low-income households in Belgium. For each sample a different mode of data collection was used (online

**Table 5. Average variance extracted (AVE) and squared inter-construct correlations (Sample 2, n = 1265).**

| Latent constructs | AVE | Latent constructs | | |
|---|---|---|---|---|
| | | Information costs | Process costs | Stigma |
| **Information costs** | 0.500 | 1.000 | | |
| **Process costs** | **0.700** | 0.114 | 1.000 | |
| **Stigma** | 0.500 | 0.000 | 0.004 | 1.000 |

and face-to-face). Results showed that a scale with nine items was relevant to measuring claiming costs and presented good psychometric properties.

In the item development phase, initial items of the instrument were generated based on a thorough review of the literature on the non-take-up of benefits. Subsequently, expert consultation was used to ensure the content validity of the scale. Items were further refined based on input from 'experts by experience in poverty and deprivation' to ensure that the items of the scale were clear, understandable and appropriate. In the scale development phase, we submitted the items to an exploratory factor analysis, which enabled us to test for the factor structure of the items and to retain a parsimonious set of items. Based on the EFA, we retained ten items grouped in three dimensions: information costs, process costs and stigma. In the scale evaluation phase, with a different sample, this three-factor structure was tested through confirmatory factor analysis. After dropping one more item, CFA demonstrated a three-factor model with nine items to be a good fit to the data. The CCS and its three subscales presented adequate to good internal consistencies. Also, the convergent and divergent validity was good both at item and subscale level.

As a first effort to develop an instrument that can be used to measure the claiming costs potential beneficiaries face when claiming benefits, this study has limitations that warrant further research. First, even though the Claiming Costs Scale was developed based on a solid theoretical background, since claiming costs can exist in various forms, other dimensions of costs besides the three key dimensions (information, process costs and stigma) may be relevant to consider. Especially with regard to stigma, our instrument only includes items that cover feelings and beliefs about personal and social stigma, whereas in the literature also a third form of stigma is highlighted that we do not cover explicitly (i.e. claims stigma or stigma that arises from the process of claiming benefits). Second, this study was conducted on samples targeted at the low-income population in Belgium, which limits the generalizability of our findings. Despite the fact that we tested for the dimensionality of the instrument in two different samples, which contributes to the validity of our instrument, it will be interesting to confirm the theoretical three-factor structure on a new sample both in Belgium and elsewhere to ascertain functional equivalence, latent factor, and factor loadings. Third, convergent and divergent validity at the scale level were not examined and future studies should include theses aspects. Fourth, test-retest reliability analysis is also required to consolidate the results. Finally, it would be interesting to evaluate the scale's performance for assessing claiming costs associated with other benefit types.

Despite these limitations, the Claiming Costs Scale proved to be a short, valid and reliable measure of claiming costs, representing a relevant addition to the existent measures. To the best of our knowledge, this is the first standardised scale which combines all three key dimensions of claiming costs into an instrument that has been systematically tested among a low-income population. While we tested the scale for social assistance benefits for able-bodied persons in Belgium, the scale's items can be easily adapted to measure claiming costs for a very broad range of social benefits and services that struggle with non-take-up. Furthermore, the items included in the scale are formulated in such a way that they can be asked to both potential beneficiaries who do not take-up social benefits and those who do take up the benefit under study. This allows for a consistent analysis of the potential role the various dimensions of claiming costs play in driving non-take-up, as well as the overall importance of claiming costs as compared to other factors that may explain the non-take-up of social benefits. If researchers are interested in understanding the complex phenomenon of non-take-up and assessing how take-up may be improved in the future, they are encouraged to include the Claiming Costs Scale in both large-scale surveys and more qualitive discussions about non-take-up. In addition, service providers who are interested in understanding the non-take-up of

their service can use the instrument in more targeted assessments to detect potential bottle-necks for their clients during the application process and implement local practices in their service delivery to reduce the costs of claiming for their clients.

## Conclusion

Better understanding why individuals do not take up the social benefits that they are entitled to is important for both researchers, policy makers and service providers. This would help to identify key changes at the level of benefit design and service delivery to improve the uptake of benefits. Therefore, we developed the Claiming Costs Scale, which measures on the basis of just nine items the practical, psychological and social costs potential beneficiaries face when claiming benefits. We discerned three subscales: information costs or the expected, perceived and experienced time and effort involved in gathering and understanding the (correct) information about the benefit and application procedure; process costs or the expected, perceived and experienced time, effort and money involved in the application process itself; and the expected, perceived and experienced stigma associated with claiming and receiving social benefits. Our instrument proved to have satisfying psychometric properties in terms of validity and reliability. Further testing of the psychometric properties of the scale is recommended by conducting studies in different populations and for different types of benefits.

## Supporting information

**S1 Appendix. Questionnaire online survey TAKE.**
(DOCX)

**S2 Appendix. Questionnaire TAKE survey.**
(DOCX)

## Acknowledgments

We would like to thank all participants for their contribution to this study, as well as the academic experts, experts by experience and professional interviewers without whose support this study would not have been finished. The authors are also thankful for the help of the colleagues received from the TAKE Project.

## Author Contributions

**Conceptualization:** Julie Janssens, Tim Goedemé.

**Formal analysis:** Julie Janssens, Koen Ponnet.

**Funding acquisition:** Tim Goedemé.

**Investigation:** Julie Janssens.

**Methodology:** Julie Janssens, Koen Ponnet.

**Project administration:** Tim Goedemé.

**Supervision:** Tim Goedemé.

**Writing – original draft:** Julie Janssens, Tim Goedemé, Koen Ponnet.

**Writing – review & editing:** Julie Janssens, Tim Goedemé, Koen Ponnet.

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
