## [Decision Letter · Decision Letter 0]

17 May 2021

PONE-D-21-04347

The Claiming Costs Scale: a new instrument for measuring the costs potential beneficiaries face when claiming social benefits.

PLOS ONE

Dear Dr. Janssens,

Thank you for submitting your manuscript to PLOS ONE. After careful consideration, we feel that it has merit but does not fully meet PLOS ONE’s publication criteria as it currently stands. Therefore, we invite you to submit a revised version of the manuscript that addresses the points raised during the review process.

We look forward to receiving your revised manuscript.

Kind regards,

Ghaffar Ali, PhD

Academic Editor

PLOS ONE

Additional Editor Comments:

Please address reviewer's comments mentioned below comprehensively and respond them one by one with reasonable changes shown separately.

Apart form that, please also adjust the following in your revised version.

1. Never put a dot in the title.

2. Do not divide abstract into three paragraphs. Abstract should be a single paragraph no more than 250 words.

3. Include ripple effects of the findings practically in the abstract and same is for conclusions.

4. Research design should be "Materials and Methods".

5. There are many concerns regarding the methodology of this paper. Many issues have been raised by the authors, some are here as following:

6. How authors reach to that sample size? and what method was used to sampling and why?

7. If correlation was used, then insert the equations and specify that which information/data/ items were considered in that.

8. In doing so, please make sure your methodology section should not be extended.

9. Discussion should be a separate section than conclusions.

10. Please provide clear conclusions and practical policy implications in a separate section.

11. Is it possible to add more Figures? 1 or 2 additional?

12. The paper is not completely formatted as prescribed by the guidelines of the journal. Please read authors Instructions section carefully and modify the whole paper into desirable format of the PLOS ONE.

Journal Requirements:

Reviewers' comments:

Reviewer's Responses to Questions

**Comments to the Author**

1. Is the manuscript technically sound, and do the data support the conclusions?

Reviewer #1: Yes

2. Has the statistical analysis been performed appropriately and rigorously? 

Reviewer #1: Yes

3. Have the authors made all data underlying the findings in their manuscript fully available?

Reviewer #1: Yes

4. Is the manuscript presented in an intelligible fashion and written in standard English?

Reviewer #1: Yes

5. Review Comments to the Author

Reviewer #1: The authors developed a new scale called Claiming Cost Scale (CCS) to measure beneficiaries' claiming costs. The generation and evaluation of the scale follow a reasonable and rigorous multi-phase procedure. The validity of the scale is confirmed at the end. Here are my comments.

1. Line 325-326: The authors said, 3 factors had eigenvalues greater than 1 and scree plot confirmed this three-factor solution. The scree plot is not available in the current paper, it would be helpful to show it. And maybe also add some simple explanation that the eigenvalue cutoff 1 is by Kaiser Rule (I suppose), otherwise it may seem like an arbitrary cutoff.

2. In Figure 1: The meanings of some numbers (for example -0.03 between Information costs and Stigma) are not very clear; also the meanings of "e" in circles on the left are not very clear. More explanation and legends would be helpful.

3. For the Confirmatory factor analysis: In line 257-258, authors said "A reasonably well-fitting model should have a statistically insignificant Chi-square" and "a relative Chi-square with a value between 2.00 and 5.00". I suppose the "relative Chi-square" here means chi-square test statistic divided by its degrees of freedom (df); if so, the p-value is not only depending on the value of relative Chi-square, but also largely on the df. For example, when df = 1, a relative chi-square of 2 has p-value 0.157; when df = 10, a relative chi-square of 2 has p-value 0.029. So what is the justification of "between 2.00 and 5.00"? Then in line 341-345, the p-values of chi-squares are both less than 0.001, which are significant. This does not mean the analysis is inadequate since it is on a second sample of size 1265 (much larger than first sample of 141) and anything could happen in real data analysis. But more discussions about this, like some potential problems or issues, are desired. (For example, the discrepancy between two samples, like in Table 1, 10% of first sample have monthly disposable income less than 499,99 EUR, while only 2% in second sample.)

4. Line 362: "loadings above .50, except for item ‘S6’", but it seems in Figure 1, S2 has factor loading 0.49, not S6.

5. Line 245-251: The authors said almost 2000 households were interviewed, 1265 individuals livings on a relatively low-income and between 18 and 64 years old were used as the second sample. The number of individuals interviewed (about 2000) is greater than size of second sample 1265, is it because individuals with relatively high income or not between 18 and 64 were excluded? And also, are these 1265 individuals independent, i.e. each household contributes at most 1 individual in the 1265 sample? (Two or more individuals from same household are dependent)

6. PLOS authors have the option to publish the peer review history of their article (what does this mean?). If published, this will include your full peer review and any attached files.

Reviewer #1: No

---

## [Author Response · Author response to Decision Letter 0]

19 Jul 2021

Review “The Claiming Costs Scale: a new instrument for measuring the costs potential beneficiaries face when claiming social benefits”

Additional Editor Comments:

Dear editor,

We would like to thank the reviewer and you for your time and the constructive comments and useful suggestions on our manuscript entitled “The Claiming Costs Scale: a new instrument for measuring the costs potential beneficiaries face when claiming social benefits.” We tried to take the helpful feedback maximally into account when making changes to the manuscript. We believe the feedback has led to a substantial improvement of the manuscript. 

Please find below our responses to the comments. We give a detailed overview of the changes we made in the current version of the manuscript. In order to indicate the revisions made, we used colored text to highlight the changes in the revised version of our manuscript. 

Thank you again for your valuable and well-appreciated comments. Please let us know if we can be of further help.

Yours sincerely,

The authors

1. Never put a dot in the title.

Authors: 

Thank you for this comment. This has been adjusted.

2. Do not divide abstract into three paragraphs. Abstract should be a single paragraph no more than 250 words.

Authors: 

We have adjusted as suggested.

3. Include ripple effects of the findings practically in the abstract and same is for conclusions.

Authors: 

This has been changed.

4. Research design should be "Materials and Methods".

Authors: 

We have adjusted this in the revised version of the paper.

5. There are many concerns regarding the methodology of this paper. Many issues have been raised by the authors, some are here as following:

6. How authors reach to that sample size? and what method was used to sampling and why?

Authors: 

We have clarified this in the revised version of the paper. On page 9, we added the following information on sample 1: 

“Respondents were recruited from both Dutch and French speaking parts of Belgium. We used a quota sampling design to ensure an equal representation of Dutch and French speaking respondents. Participants were recruited in collaboration with the digital data collection provider “Research Now”, from their market research panels. All Research Now research activities are based on informed consent ("opt-in"). Once new panel members register, the information provided at that point is very clear on the purpose of their membership and they are required to complete the panel registration and agree to all terms and conditions upon registration. In addition, each invited panel member was free to participate in our study and could simply refuse not to participate to the online survey that was sent to them by e-mail.”

On page 12, we added the following information on sample 2: 

“To select the sample for this survey, we used a two-stage stratified sample design with random sampling at both stages. At the first stage a sample of municipalities was selected, with probabilities of selection proportional to estimated size, stratified according to geographical characteristics and the degree to which persons entitled to social assistance are overrepresented in the population. At the second stage, a sample of households was selected without replacement for each selected municipality, stratified by age and social assistance status. The sampling frame consisted of national register data, enriched with data on income and benefit receipt from the tax and social security registers. Next, sample members were sent a letter with an explanation about the study and an invitation to participate. Sample members were assured they could refuse at any time to participate in the study. Respondents that agreed to participate, were interviewed by trained interviewers by means of computer-assisted face-to-face interviews. At the start of the interview, respondents were asked to give their informed consent to participate in the survey by means of a registered question at the beginning of the questionnaire. If the respondent refused to agree with the conditions of the study at this specific moment, the interview was immediately interrupted.”

Furthermore, on page 13 of the revised paper, we have added the following information: 

“For this study, we made use of the information collected from the household head, who provided detailed information on their knowledge, experience and receipt of benefits, their financial situation, sociodemographic characteristics, household composition, housing situation and competences. Respondents who indicated to know the social assistance benefit for persons at active age and were between 18 and 64 years old, were asked to respond to the items of the Claiming Costs Scale. In total, 1,265 household heads responded to the items of the Claiming Costs Scale. The descriptive characteristics of this group (Sample 2) can be found in Table 1. The particular questions used for this study are translated into English and provided in Appendix S2.”

7. If correlation was used, then insert the equations and specify that which information/data/ items were considered in that.

Authors: 

We followed your advice and have added the following information on page 18 of the revised version of the paper: 

“Only the correlation between the latent factor ‘information costs’ and ‘process costs’ was positively significant, indicating that a higher degree of information costs is also associated with a higher degree of process costs. In contrast, we did not find a significant relation between the degree of stigma and degree of information or process costs. Fig 2 shows the model with the standardized factor loadings of the items and correlation coefficients between the latent factors.”

8. In doing so, please make sure your methodology section should not be extended.

Authors: 

We have tried to add the clarifications with as little additions to the text as possible.

9. Discussion should be a separate section than conclusions.

Authors: 

We have made two different sections for discussion and conclusions.

10. Please provide clear conclusions and practical policy implications in a separate section.

Authors: 

We have now a separate section which summarizes the practical policy implications.

11. Is it possible to add more Figures? 1 or 2 additional?

Authors: 

We have added a Figure on page 17 of the revised paper: “Fig 1. Scree plot for determining factors of the designed instrument (Sample 1, n=141)”

12. The paper is not completely formatted as prescribed by the guidelines of the journal. Please read authors Instructions section carefully and modify the whole paper into desirable format of the PLOS ONE.

Authors: 

In the revised version, we followed the formatting guidelines.

Journal Requirements:

Authors: 

Done.

Authors: 

We have now made both questionnaires available online.

Authors: 

We have added a more extensive discussion of this on page 12 of the revised paper:

“Sample members were sent a letter with an explanation about the study and an invitation to participate. Sample members were assured they could refuse at any time to participate in the study. Respondents that agreed to participate, were interviewed by trained interviewers by means of computer-assisted face-to-face interviews. At the start of the interview, respondents were asked to give their informed consent to participate in the survey by means of a registered question at the beginning of the questionnaire. If the respondent refused to agree with the conditions of the study at this specific moment, the interview was immediately interrupted.”

Authors: 

The dataset was part of a larger data authorisation agreed by the Data Protection Authority of Belgium. We have now singled out the variables that we used for this article, which we have made available online in pseudonomynised format, together with essential metadata. This should allow others to replicate the study without any problem.

Authors: 

We now have made the relevant de-identified data sets that we used for this article publicly available. 

Authors: 

We have now uploaded the necessary data for replication for both samples that we use in this study and posted them in the following repository: https://doi.org/10.3886/E131962V3

Authors: 

Done. We provide the following information on page 7:

“The study protocol was approved by the Ethics Committee for Social Sciences and Humanities of [name blinded for peer review] and informed consent was obtained from all participants before completing the questionnaire. All data were analysed anonymously, and cannot be linked back to identifying information.”

Reviewers' comments:

Reviewer's Responses to Questions

Comments to the Author

1. Is the manuscript technically sound, and do the data support the conclusions?

Reviewer #1: Yes

2. Has the statistical analysis been performed appropriately and rigorously? 

Reviewer #1: Yes

3. Have the authors made all data underlying the findings in their manuscript fully available?

Reviewer #1: Yes

4. Is the manuscript presented in an intelligible fashion and written in standard English?

Reviewer #1: Yes

5. Review Comments to the Author

Reviewer #1: The authors developed a new scale called Claiming Cost Scale (CCS) to measure beneficiaries' claiming costs. The generation and evaluation of the scale follow a reasonable and rigorous multi-phase procedure. The validity of the scale is confirmed at the end. Here are my comments.

Authors: 

We would like to thank the reviewer for the constructive comments and suggestions, which we have incorporated in the revised text.

1. Line 325-326: The authors said, 3 factors had eigenvalues greater than 1 and scree plot confirmed this three-factor solution. The scree plot is not available in the current paper, it would be helpful to show it. And maybe also add some simple explanation that the eigenvalue cutoff 1 is by Kaiser Rule (I suppose), otherwise it may seem like an arbitrary cutoff.

Authors: 

We have added a Figure on page 17 of the revised paper: “Fig 1. Scree plot for determining factors of the designed instrument (Sample 1, n=141)”

On page 10 of the revised paper, we provide some more information on the criteria we use to extract the main factors and withhold the indicators of the factors: “ We performed an EFA using principal axis factoring and varimax rotation to extract the main factors. Any factor with an eigenvalue above 1 was considered significant for factor extraction, and a scree plot was used to specify the number of factors. Factor loadings equal to or greater than .40 were considered acceptable [52], however factor loadings greater than .50 were preferred [53, 54].

2. In Figure 1: The meanings of some numbers (for example -0.03 between Information costs and Stigma) are not very clear; also the meanings of "e" in circles on the left are not very clear. More explanation and legends would be helpful.

Authors: 

Thank you for this comment. We have now explained the figure better in the text, making a clear distinction between factor loadings and correlations between latent factors. We also added an explanation of the correlation between the latent factors on page 18 of the revised paper: 

“Only the correlation between the latent factor ‘information costs’ and ‘process costs’ was positively significant, indicating that a higher degree of information costs is also associated with a higher degree of process costs. In contrast, we did not find a significant relation between the degree of stigma and degree of information or process costs. Fig 2 shows the model with the standardized factor loadings of the items and correlation coefficients between the latent factors.”

In addition, we replaced the initial figure of the 3-factor model obtained from the confirmatory factor analysis with a new figure where the standard errors (“e”) of the items are now shown.

3. For the Confirmatory factor analysis: In line 257-258, authors said "A reasonably well-fitting model should have a statistically insignificant Chi-square" and "a relative Chi-square with a value between 2.00 and 5.00". I suppose the "relative Chi-square" here means chi-square test statistic divided by its degrees of freedom (df); if so, the p-value is not only depending on the value of relative Chi-square, but also largely on the df. For example, when df = 1, a relative chi-square of 2 has p-value 0.157; when df = 10, a relative chi-square of 2 has p-value 0.029. So what is the justification of "between 2.00 and 5.00"? Then in line 341-345, the p-values of chi-squares are both less than 0.001, which are significant. This does not mean the analysis is inadequate since it is on a second sample of size 1265 (much larger than first sample of 141) and anything could happen in real data analysis. But more discussions about this, like some potential problems or issues, are desired. (For example, the discrepancy between two samples, like in Table 1, 10% of first sample have monthly disposable income less than 499,99 EUR, while only 2% in second sample.)

Authors: 

On page 13 of the revised paper, we explain more in detail the use of the relative Chi-square and the benchmark we use to evaluate our models based on the value of the relative Chi-square. In addition, we also explain why having a significant value for Chi-square is not necessarily problematic in larger sample sizes:

“We use the relative Chi-square, which equals the Chi-square value divided by the degrees of freedom, as this value might be less sensitive to sample size then the classic Chi-square index. A reasonably well-fitting model should have a statistically insignificant Chi-square, however in larger samples (n>200) the model will be routinely rejected based on the Chi-square value. Despite the fact there is still some discussion about the value the relative Chi-square should have (for an overview see [59]), most researchers recommend to have a value of 5 or less.”

4. Line 362: "loadings above .50, except for item ‘S6’", but it seems in Figure 1, S2 has factor loading 0.49, not S6.

Authors: 

Many thanks for pointing out this error. This has now been corrected.

5. Line 245-251: The authors said almost 2000 households were interviewed, 1265 individuals livings on a relatively low-income and between 18 and 64 years old were used as the second sample. The number of individuals interviewed (about 2000) is greater than size of second sample 1265, is it because individuals with relatively high income or not between 18 and 64 were excluded? And also, are these 1265 individuals independent, i.e. each household contributes at most 1 individual in the 1265 sample? (Two or more individuals from same household are dependent)

Authors: 

We have now clarified in the text that this is on the basis of one interview per household. We have also clarified that the sample used in the study is smaller than the original sample because of the age restriction (primary reason) and the requirement of knowing the social assistance benefit (minor reason for exclusion from the analysis). We included this information on page 13 of the revised text:

“In total, almost 2000 households living in the three regions of Belgium were interviewed. For this study, we made use of the information collected from the household head, who provided detailed information on their knowledge, experience and receipt of benefits, their financial situation, sociodemographic characteristics, household composition, housing situation and competences. Respondents who indicated to know the social assistance benefit for persons at active age and were between 18 and 64 years old, were asked to respond to the items of the Claiming Costs Scale. In total, 1,265 household heads responded to the items of the Claiming Costs Scale.”

---

## [Decision Letter · Decision Letter 1]

9 Aug 2021

The Claiming Costs Scale: a new instrument for measuring the costs potential beneficiaries face when claiming social benefits

PONE-D-21-04347R1

Dear Dr. Janssens,

We’re pleased to inform you that your manuscript has been judged scientifically suitable for publication and will be formally accepted for publication once it meets all outstanding technical requirements.

Kind regards,

Ghaffar Ali, PhD

Academic Editor

PLOS ONE

Additional Editor Comments (optional):

Reviewers' comments:

Reviewer's Responses to Questions

**Comments to the Author**

1. If the authors have adequately addressed your comments raised in a previous round of review and you feel that this manuscript is now acceptable for publication, you may indicate that here to bypass the “Comments to the Author” section, enter your conflict of interest statement in the “Confidential to Editor” section, and submit your "Accept" recommendation.

Reviewer #1: All comments have been addressed

2. Is the manuscript technically sound, and do the data support the conclusions?

Reviewer #1: (No Response)

3. Has the statistical analysis been performed appropriately and rigorously? 

Reviewer #1: (No Response)

4. Have the authors made all data underlying the findings in their manuscript fully available?

Reviewer #1: (No Response)

5. Is the manuscript presented in an intelligible fashion and written in standard English?

Reviewer #1: (No Response)

6. Review Comments to the Author

Reviewer #1: (No Response)

7. PLOS authors have the option to publish the peer review history of their article (what does this mean?). If published, this will include your full peer review and any attached files.

Reviewer #1: No

---

## [Editor Report · Acceptance letter]

12 Aug 2021

PONE-D-21-04347R1 

The Claiming Costs Scale: a new instrument for measuring the costs potential beneficiaries face when claiming social benefits 

Dear Dr. Janssens:

I'm pleased to inform you that your manuscript has been deemed suitable for publication in PLOS ONE. Congratulations! Your manuscript is now with our production department. 

Kind regards, 

on behalf of

Prof. Ghaffar Ali 

Academic Editor

PLOS ONE